# MINICTX: NEURAL THEOREM PROVING WITH (LONG-) CONTEXTS

**Jiewen Hu     Thomas Zhu     Sean Welleck**
Carnegie Mellon University

## ABSTRACT

Real-world formal theorem proving often depends on a wealth of context, including definitions, lemmas, comments, file structure, and other information. We introduce `miniCTX`, which tests a model's ability to prove formal mathematical theorems that depend on new context that is not seen during training. `miniCTX` contains theorems sourced from real Lean projects and textbooks, each associated with a context that can span tens of thousands of tokens. Models are tasked with proving a theorem given access to code from the theorem's repository, which contains context that is needed for the proof. As a baseline for `miniCTX`, we tested fine-tuning and prompting methods that condition theorem proving on preceding context. Both approaches substantially outperform traditional methods that rely solely on state information. We found that this ability to use context is not captured by previous benchmarks such as `miniF2F`. Alongside `miniCTX`, we offer NTP-TOOLKIT for automatically extracting and annotating theorem proving data, making it easy to add new projects into `miniCTX` to ensure that contexts are not seen during training. `miniCTX` offers a challenging and realistic evaluation of neural theorem provers.[1]

## 1 INTRODUCTION

Formal theorem proving in interactive theorem provers (ITPs) provides a testbed for evaluating the reasoning capabilities of large language models (LLMs). Theorem proving capabilities can then directly translate to automation for mathematicians, such as via tools that complete or formalize proofs (Welleck & Saha, 2023; Song et al., 2024; Welleck, 2024; Agrawal et al., 2022). However, despite their promise, we see a gap between the evaluation of current language model-based provers and the complexity of real-world theorem proving.

Our motivating observation is that theorems and proofs depend on various forms of *context*, such as newly-defined definitions and lemmas. For instance, to prove results about a square, one might first formalize a definition of a rectangle, prove some results about rectangles, then specialize them to a newly-defined square (Kontorovich, 2024b) (Figure 1). However, existing methods for training and evaluating LLM-based theorem provers often fail to incorporate the full range of contextual information available in real-world projects. For example, benchmarks often focus on proving standalone competition problems (e.g., `miniF2F` (Zheng et al., 2022)) or theorems from a library that the model has trained on (e.g., Mathlib (Han et al., 2022; Yang et al., 2023)), and state-of-the-art LLM-based provers are trained to accept only a proof state as input, making them unaware of new theorems and definitions (Polu & Sutskever, 2020; Ying et al., 2024; Xin et al., 2024). While some existing work, including premise selection techniques (Jiang et al., 2022; Mikuła et al., 2023; Yang et al., 2023) and datasets like CoqGym (Yang & Deng, 2019), have explored theorem proving based on information beyond the current state, they often focus only on providing relevant premises—lemmas that can assist proof construction—which are only a subset of the available information.

Building on these foundations, we propose `miniCTX`: a benchmark that seeks to expand the scope of context used in theorem proving. We extend beyond traditional premise selection explored in prior benchmarks (e.g., Yang et al. (2023); Yang & Deng (2019)) by incorporating a more comprehensive set of contextual elements. This includes premises, prior proofs, comments, notation, and structural

---

[1] Project page: `https://cmu-l3.github.io/minictx`. Please refer to our project page for our dataset and evaluation links, and future updates including `miniCTX-v2`.

Table 1: Comparison of theorem proving benchmarks across several key features.

| Benchmark | Language | Premise | Full Context | Multi-source | Temporal Split |
|---|---|---|---|---|---|
| miniF2F (Zheng et al., 2022) | Multiple | ✗ | ✗ | ✗ | ✗ |
| ProofNet (Azerbayev et al., 2023) | Lean | ✗ | ✓ | ✓ | ✗ |
| LeanDojo (Yang et al., 2023) | Lean | ✓ | ✗ | ✗ | ✗ |
| LeanStep (Han et al., 2022) | Lean | ✓ | ✗ | ✓ | ✗ |
| CoqGym (Yang & Deng, 2019) | Coq | ✓ | ✗ | ✓ | ✗ |
| PISA (Jiang et al., 2021) | Isabelle | ✗ | ✗ | ✓ | ✗ |
| miniCTX (Ours) | Lean | ✓ | ✓ | ✓ | ✓ |

components like imports and declarations. By doing so, `miniCTX` aims to drive the development of methods that understand and work with context that occurs in complex, real-world theorem proving tasks. We compare `miniCTX` with several popular theorem proving datasets to highlight its unique contributions in terms of contextual dependency and real-world applicability (see Table 1). Additionally, considering the common use of pre-trained language models we mitigate potential data contamination by continually and automatically updating `miniCTX` with new Lean projects, so that evaluated theorems are not seen during training. Our key contributions are:

`miniCTX` **Benchmark:** We introduce `miniCTX`, the first benchmark designed specifically to evaluate theorem proving in real-world settings where proofs depend on in-file definitions, lemmas, and context from formal projects. `miniCTX` presents a unique challenge by requiring models to reason over long contexts and handle dependencies that arise in real-world theorem proving tasks.

**NTP-TOOLKIT:** To facilitate the automatic updating of `miniCTX`, we developed the NTP-TOOLKIT, which automatically extracts relevant theorems and contexts from Lean projects. Additionally, we provide a Lean REPL wrapper that enables simpler evaluation on `miniCTX`.

**Baseline Evaluations:** We evaluate `miniCTX` on several existing baseline models, including different fine-tuning and prompting strategies, as well as methods with premise selection. We also propose file-tuning, a strong baseline method for training models using full file contexts, where both the theorem statements and their surrounding context are provided during training. This approach establishes a robust baseline for future work on context-dependent theorem proving.

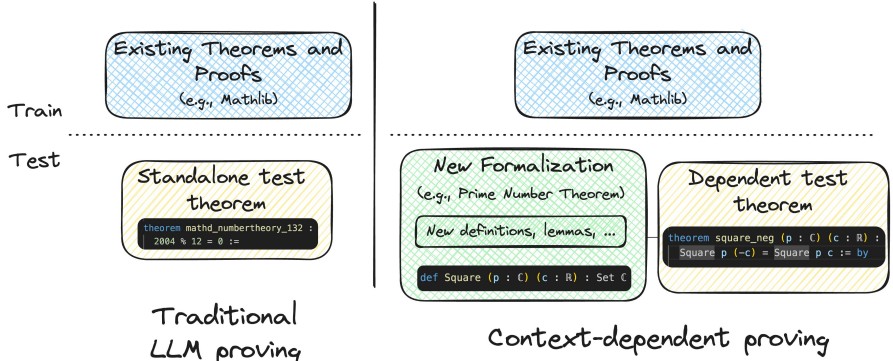

Figure 1: Many state of the art provers are trained on a static dataset of theorems and proofs, then evaluated on standalone problems such as competition problems (left). We argue that neural provers must also operate in the realistic *context-dependent* setting, in which results depend on working with new mathematical objects and their facts, notations, and the structural elements of the project (imports, variables, etc.) (right).

## 2 THEOREM PROVING WITH CONTEXT

For language model-based provers to function as useful tools in this real-world setting, they need to be able to work with new information such as new definitions or lemmas. For example, a system suggesting proofs in the Prime Number Theorem project should be familiar with the project's

definition of "square". Many current language model-based provers are trained on a static snapshot of data, and are hence unaware of any context that was created after the model was trained (Figure 1). They are often evaluated on either context-less standalone competition problems (Zheng et al., 2022; Azerbayev et al., 2023) that do not reflect realistic settings, or existing Mathlib proofs (Yang et al., 2023) which risks data contamination. As a result, it remains unclear whether these models can work with new information, which is necessary for using them as an assistant in a real proof development, and how to enable this capability.

**Context-dependent proving.** We study *context-dependent theorem proving*, where the goal is for a model to generate proofs $y$ for new theorems $x$, based on a context $c$ that includes background information such as definitions, lemmas, or natural language comments. Formally, the problem is

$$\text{maximize}_M \ \mathbb{E}_{(x,c)\sim p}\mathbb{E}_{y\sim M(\cdot|x,c)} v(x, c, y), \tag{1}$$

where $(x, c) \sim p$ denotes a (theorem, context) pair from a context distribution $p$, $M$ is a model that produces a proof $y$, and $v$ returns 1 if the proof is correct and 0 otherwise.

**Evaluating context-dependent proving.** We choose Lean (Moura & Ullrich, 2021) as the verifier $v$, because of the large body of recent theorems in Lean that can be used as evaluation data, and the abundance of proving methods in Lean that we use as baselines. We treat a Lean repository as the distribution $p$. Each context $c$ is a subset of the repository, including new definitions, lemmas, notations, imports, and comments that are relevant to the theorem.

Given a language model, we can test three kinds of generalization by ensuring the following:

- *Theorem-level generalization*: the proof must not occur in the model's training data.
- *Context-level generalization*: the code $c$ and proof must not occur in the training data.
- *Project-level generalization*: the entire repository must not occur in the training data.

For baseline evaluations, we investigate the *in-file context* case where $c$ is the source code that precedes the theorem $x$ in a file, as well as *cross-file context* where $c$ includes both preceding code and relevant premises retrieved from imported modules through premise selection.

## 3  miniCTX: A BENCHMARK FOR THEOREM PROVING WITH CONTEXT

We develop miniCTX, a Lean 4 theorem proving benchmark of theorems that depend on newly-defined lemmas, definitions, and proofs from within a project. miniCTX is currently based on 762 theorems from six projects: (1) Prime Number Theorem (**PNT**) (Kontorovich, 2024a), (2) Polynomial Freiman-Ruzsa Conjecture and its extension with high cross-file dependency (**PFR**) (Tao, 2023), (3) recent results from the standard mathematical library (**Mathlib**), (4) an introductory text on theorem proving (**HTPI**) (Macbeth, 2023), (5) high energy physics formalization in HepLean (**HEP**) (Tooby-Smith, 2024), and (6) scientific computing formalizations (**SciLean**) (Skřivan, 2021). These theorems are equally split into 381 validation and 381 test theorems. Table 2 shows an overview of the dataset. Each theorem in miniCTX consists of the theorem statement, preceding file contents up to the theorem statement, and metadata, in JSON (see §A.1).

1. Theorem statement,
2. Preceding file contents up to the theorem statement,
3. Metadata, including:
    (a) File name,
    (b) Project commit and version,
    (c) Commit and time at which the theorem and its file was added,
    (d) Position of the theorem in the file and number of premises preceding it,
    (e) Number of in-file premises and cross-file premises used by the statement or proof,
    (f) Imported modules (for premise selection support),
    (g) Proof length and type.

Using our benchmark, users can easily reconstruct the complete context for each theorem, including both in-file and cross-file context. The in-file context is provided directly by preceding file contents, while the cross-file context can be reconstructed using the metadata, which includes information on imported modules. We open-source the dataset and evaluation code.

Table 2: Problem statistics in `miniF2F` (Zheng et al., 2022) and `miniCTX`.

|  | Split | Problems valid + test | Context Size (tokens) | In-File Premises (premises / 100 tokens) | Repo. Premises (premises / 100 tokens) | Proof Size (lines) |
|---|---|---|---|---|---|---|
| `miniF2F` |  | 244 + 244 | 153* | — | — | 3.0[†] |
| `miniCTX` | PNT | 85 + 85 | 10,858 | 1.87 | 0.30 | 3.3 |
|  | PFR | 51 + 51 | 18,059 | 0.65 | 1.10 | 27.2 |
|  | PFR$_{cross}$ | 43 + 43 | 4,351 | 0.44 | 2.75 | 2.7 |
|  | Mathlib | 50 + 50 | 14,440 | — | — | 6.1 |
|  | HTPI | 45 + 45 | 65,082 | 2.85 | 0.00 | 10.7[†] |
|  | HEP | 61 + 61 | 3,585 | 5.65 | 4.25 | 3.1 |
|  | SciLean | 46 + 46 | 6,249 | 2.08 | 9.72 | 1.8 |
|  | **All** | 381 + 381 | 18,690 | 1.94 | 2.63 | 8.5 |

*Only counting library imports and definitions.  [†]Excluding theorems without proofs.

## 3.1 miniCTX Sources

**PNT.** PrimeNumberTheoremAnd (Kontorovich, 2024a) is a project started in January 2024 that formalizes the prime number theorem in Lean as well as related concepts, such as counter integral on rectangles in $\mathbb{C}$. We find the files `Rectangle.lean` and `ResidueCalcOnRectangles.lean` suitable for our purpose of testing context-dependent theorem proving, especially when we use preceding file content as context, as each file is self-contained within the project and contains new definitions (rectangles, squares) and many interdependent lemmas. See §A.2 for an illustration of such lemmas.

**PFR.** PFR (Tao, 2023) is a project started in November 2023 that formalizes a proof of the Polynomial Freiman–Ruzsa (PFR) conjecture. We included 51 validation and 51 test theorems from PFR. We find that proofs of theorems in PFR tend to be much more monolithic and longer in length than those in Mathlib or other libraries. PFR also defines custom mathematical concepts and notations (such as Ruzsa distance) and a proof typically depends on many lemmas in PFR outside the current file.

**PFR$_{crossfile}$.** PFR$_{crossfile}$ is an extension of the PFR split, which includes additional problems to further evaluate cross-file dependencies. To evaluate models' performance on problems with extensive cross-file dependencies, we added 43 test theorems from `Entropy.Group` and `Entropy.Kernel.Group`, which have the highest cross-file dependencies in the project. These problems contain three times the number of cross-file premises compared to other math splits, making them a strong candidate for evaluating a model's ability to utilize cross-file premises. 43 validation theorems are chosen similarly.

**Recent Mathlib Commits.** Mathlib (Mathlib Community, 2020), is Lean's largest community-maintained repository, encompassing a wide range of mathematical concepts, programming APIs, and common tactics. It is commonly used for training theorem-proving models and is frequently updated with new definitions, theorems, and refactorings. Therefore, to avoid data contamination, we included 50 test and 50 validation theorems newly added to Mathlib since March 2024, by filtering recent Mathlib commits to ones that only add new theorems. Assuming that the model was trained prior to April 2024, the Mathlib split guarantees the evaluation of theorem-level generalization.

**HTPI.** HTPI contains the Lean code for the book *How to Prove It* (HTPI) (Velleman, 2019), which explains a systematic approach to constructing mathematical proofs with Lean. It covers topics like elementary logic and number theory, and proving techniques like induction.The files in HTPI typically start with basic definitions and lemmas that might be used throughout the entire file, followed by exercises and several example problems. Therefore, models can utilize definitions, lemmas, and proof structures from example problems to solve exercises, making it an effective benchmark for testing context-dependent theorem-proving models.

**HEP.** HepLean (Tooby-Smith, 2024) is an open-source project that digitalizes definitions, theorems, proofs, and calculations in high energy physics using Lean. HepLean aims to facilitate discovery, automate new findings, verify correctness, and provide educational tools in physics. We selected files from the space and time section, which introduces several new definitions, including 4D Euclidean spacetime models. These files contain high in-file and cross-file dependencies, ideal for evaluating context-dependent theorem proving. This split represents an area outside of mathematics, further expanding the generalization of the benchmark.

**Recent SciLean Commits.** SciLean (Skřivan, 2021) is an open-source project designed to formalize concepts in scientific computing using Lean 4. SciLean aims to provide formalized tools and frameworks for efficiently representing and proving properties of numerical methods, differential equations, and optimization problems, bridging applied mathematics and computer science. Given Scilean has been created for 3 years and accessed by multiple projects, similar to our Mathlib split, we selected 46 test and 46 validation theorems added to SciLean since March 2024 in order to ensure the data is more likely unseen for evaluating models. The newly added problems are mostly supplements to existing files, so they evaluate the model's ability to apply and learn from previous lemmas.

### 3.2 PROBLEM SELECTION METHODOLOGY

The problem selection process for different splits in `miniCTX` follows three main criteria: (1) ensuring new problems are less likely to have appeared in training data, (2) utilizing an automated selection process, and (3) promoting generalization. Depending on the properties and goals of each split, we adopted three primary approaches:

1. Recency-based selection: For popular libraries such as **Mathlib** and **SciLean**, which are highly likely to be used for training, we aim to select newly added theorems. This is achieved by sorting theorems based on the timestamp of when the theorem was first added to the project, which is extracted by NTP-TOOLKIT. This helps mitigate data contamination.

2. Random selection: For more recent projects, such as the **PFR** split, where the risk of data contamination is lower, we randomly select proved theorems to ensure a representative sample of the entire project. This approach maintains generality of the selected problems.

3. Dependency-based selection: To explicitly evaluate models' performance in context-dependent proving, we selected files based on the in-file and cross-file premise labels available in our benchmark. For the **PNT** split, we chose the file with the highest number of in-file dependencies, while for **PFR$_{\text{crossfile}}$**, we selected files with the highest cross-file dependencies. For **HEP**, we selected files with a balance of both types. The level of dependency is also extracted automatically by NTP-TOOLKIT.

Although human inspection and expertise are involved in ensuring that the selected problems are valid and sufficiently general for evaluating models across diverse settings, all selection processes are ultimately automated by using the labels extracted through our toolkit. This ensures consistency across the benchmark and scalability to future updates.

### 3.3 KEY FEATURES AND CHALLENGES

`miniCTX` introduces several key features that distinguish it from other theorem proving benchmarks, addressing challenges that have not been tackled by previous benchmarks:

**Real-world theorem proving.** Unlike popular benchmarks (e.g., miniF2F (Zheng et al., 2022), ProofNet (Azerbayev et al., 2023), FIMO (Liu et al., 2023)) that focus on isolated competition problems, real-world research-level theorem proving is heavily dependent on rich mathematical contexts. Therefore, `miniCTX` includes real-world, complex theorems from a variety of ongoing Lean projects, such as Prime Number Theorem (PNT) and Polynomial Freiman–Ruzsa Conjecture (PFR). They rigorously test a model's ability in real-world formalization projects. This diversity contrasts with the LeanDojo benchmark (Yang et al., 2023), which focuses solely on Mathlib, enabling `miniCTX` to better test a model's generalization in different settings.

**Contextual evaluation.** Proving a theorem often depends on new definitions, lemmas, or other contextual information, which a model may not have seen during training. `miniCTX` includes theorems along with this new context. During evaluation, the model is expected to leverage the provided new context to help prove the theorem.

Beyond previous datasets like LeanDojo (Yang et al., 2023) and CoqGym (Yang & Deng, 2019), which include relevant definitions and theorems, `miniCTX` includes additional useful contextual information that may make some theorems *easier* to prove compared to standalone theorems. For instance, Lean source code can have natural language comments that may help constrain the space of possible proofs. Moreover, some proofs within a file often have analogous patterns or structure, which

may make subsequent theorems easier to prove (see §A.2). These additional forms of context occur in the real-world process of formalization, yet their use in neural theorem proving is underexplored.

**Automatically updating the benchmark.** Most modern neural theorem provers use a large language model as a backbone. Therefore, it is crucial to ensure that evaluation content is not seen during (pre-)training, a problem not addressed by previous benchmarks (see §G). We plan to update `miniCTX` periodically to include theorems beyond a certain cut-off date (see our project page). Future updates will be automatically extracted from new Lean projects using NTP-TOOLKIT (§3.4). See Figure 2.

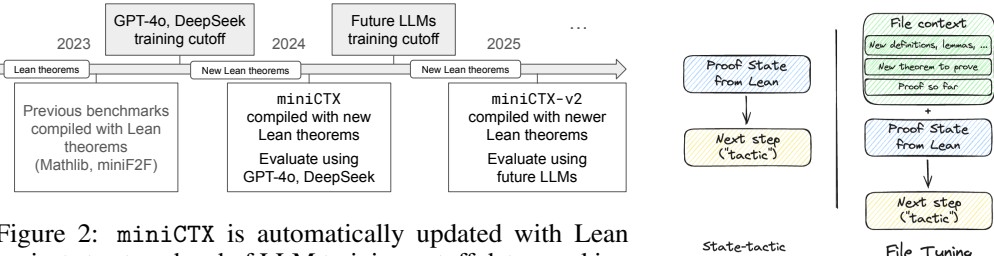

Figure 2: `miniCTX` is automatically updated with Lean projects to stay ahead of LLM training cutoff dates, making it a suitable benchmark for real-world theorem proving for pre-trained models.

Figure 3: State-tactic vs. file tuning.

### 3.4 NTP-TOOLKIT: AUTOMATED DATA EXTRACTION AND EVALUATION

We provide NTP-TOOLKIT that automatically extracts `miniCTX` and metadata. NTP-TOOLKIT is also used to extract data for training baselines, and it provides an interactive Lean evaluation environment.

**Data extraction.** NTP-TOOLKIT contains a general-purpose data extraction tool that extracts examples from an arbitrary Lean 4 repository and formats them into individual theorems in `miniCTX`. The tool is implemented in Lean based on `lean-training-data` (Morrison, 2023). Specifically, NTP-TOOLKIT takes in a configuration file with some Lean repositories specified. Then for each theorem in each repository, it outputs a JSON-formatted entry in `miniCTX`, including the full context information, theorem statement, and proof. It also extracts useful metadata that is used for automated problem selection in `miniCTX` (see §3.2), and for our experimental analysis. See Appendix §A and §C.1 for the formats provided by the data extraction.

**Interaction via Lean REPL.** In order to evaluate a model on `miniCTX`, a model needs to efficiently communicate generated proofs with Lean and receive current state and error information. To better integrate LLMs and Lean, we developed a Python wrapper for the Lean Read-Eval-Print Loop (REPL) (Lean Prover Community, 2024). The Lean REPL offers an interactive environment for submitting code to and receiving messages from a Lean instance. We provide a Python interface for (1) submitting both complete Lean proofs and individual tactics, and (2) receiving and handling feedback from Lean, such as error messages or indicators that a tactic application is valid or a proof is complete. This interface enables simpler evaluation on the `miniCTX` benchmark.

## 4 EXPERIMENTS

We evaluate several baselines on `miniCTX`, demonstrating the importance of context in real-world theorem proving. We study performance along various axes, including premise dependency, context length, difficulty, and the type of information in the context. Our experiments show that file-tuned models, which can utilize context at evaluation time, improve drastically over traditional state-tactic models in the context dependent setting. Moreover, we discover that this real-world performance boost cannot be readily measured by existing benchmarks such as `miniF2F`. Our investigation reveals several open challenges that we discuss in §5.

**Training data.** We ran NTP-TOOLKIT's next-tactic extraction on a 2023 snapshot of Mathlib, yielding 307,049 examples. We then ran NTP-TOOLKIT's instruction tuning script on these examples, yielding training examples and for state-tactic and file-tuning (see §4.1). For the file-tuning examples, as an initial method for handling the long Lean files, we either truncate the middle of an input file so that the file contents is 1024 tokens, or take only the preceding 1024 tokens, with the strategy selected

at random for each example. We open-source our training data for both file-tuning and state-tactic tuning, split into 583k train, 15k dev, and 15k test examples.

## 4.1 BASELINES

We select several baselines that we evaluate on `miniCTX`, as follows:

**Prompting LLMs.** We first test the ability of a state-of-the-art API-based model, GPT-4o, to generate the complete proof given a theorem statement in Lean, with several few-shot examples provided for guidance. We generate 8 such proof samples and measure pass@8 proof rate. We also test whether adding context in the form of preceding file contents or retrieved premises improves the proof rate.

**State-tactic prompting.** Another common approach to theorem proving using language models is to let the model generate a tactic given the current proof state (Han et al., 2022; Yang et al., 2023; Polu & Sutskever, 2020; Lample et al., 2022). Therefore, we test the *state-tactic prompting* setting, which prompts a model specialized for mathematical tasks, Llemma-7b (Azerbayev et al., 2024), to output a tactic given a proof state. At test time, the model generates one tactic at a time, and we use a best-first search to construct full proofs (Han et al., 2022; Yang et al., 2023; Polu & Sutskever, 2020).

**State-tactic tuning.** We can further improve tactic generation by fine-tuning language models on human-written (state $x_t$, tactic $y_t$) pairs. We follow this *state-tactic* framework and fine-tune a *state-tactic tuned* model from DeepSeek-Coder-1.3b (Guo et al., 2024) to input proof states and output tactics, trained on human-written tactics in Mathlib, the main mathematical library in Lean, extracted by NTP-TOOLKIT. Similarly, we use best-first search at test time.

**File-tuning.** A drawback to the setups above is that they only consider static (state, tactic) pairs, and do not take into account new context, including comments, definitions, lemmas, and file structure, encountered during test time. Therefore, we test whether supplying context $c$, in the form of preceding file contents, to the model improves performance. Similar to state-tactic tuning, we fine-tune DeepSeek-Coder-1.3b on human-written (state $x_t$, context $c$, tactic $y_t$) triples to generate tactics based on the current context and proof state, resulting in the *file-tuned* model (see Figure 3).

**Premise selection.** While in-file context offers valuable information to models, it still overlooks external resources like imported modules, which are often crucial in real-world interactive theorem proving. To better simulate a complete context and evaluate on project-level generalization, we apply premise selection to extract relevant premises from imported files within the same repository. Specifically, we use the premise retriever provided by LeanDojo (Yang et al., 2023) to identify the top 20 most relevant definitions or lemmas from imported modules and append them to the in-file context. Given that the models we are evaluating are trained on Mathlib, and they exhibit a strong ability to apply Mathlib lemmas, we did not include Mathlib in the potential premises to ensure that the models gain as much information as possible from the provided context (both infile and crossfile). All potential premises are automatically extracted using our toolkit, ensuring an efficient and automated process.

Table 3: Performance comparison (%) of different models on `miniF2F-test` and `miniCTX-test`.

| Method | miniF2F | miniCTX-test | | | | | | | |
|---|---|---|---|---|---|---|---|---|---|
| | **Test** | **Prime** | **PFR** | **PFR$_{cross}$** | **Mathlib** | **HTPI** | **HEP** | **SciLean** | **Avg.** |
| GPT-4o (full proof) | 13.52 | 7.06 | 1.85 | 6.98 | 14.00 | 13.33 | 31.15 | 6.52 | 11.72 |
| + context | — | 31.76 | 5.56 | 34.88 | 26.00 | **17.78** | 49.18 | 17.39 | 27.08 |
| + context + premise | — | 29.41 | 7.41 | 39.53 | — | 15.56 | 44.26 | 21.74 | 26.82 |
| State-tactic prompting | 28.28 | 20.00 | 5.56 | 0.00 | 16.00 | 0.00 | 31.15 | 19.57 | 14.58 |
| State-tactic tuning | 32.79 | 17.65 | 5.56 | 0.00 | 22.00 | 11.11 | 52.46 | 19.57 | 19.53 |
| File tuning | **33.61** | 40.00 | 5.56 | **44.19** | **34.00** | 15.56 | **60.66** | **45.65** | **35.94** |
| + premise | — | **42.35** | **11.11** | 16.28 | — | 8.89 | 50.82 | 32.61 | 30.21 |

## 4.2 RESULTS

**Context-dependent methods improve theorem proving.** Table 3 shows baseline performances on `miniCTX`. We see a dramatic improvement for the file-tuned model (trained on full file context) over the state-tactic model (trained only on proof states) (35.94% vs. 19.53%). Similarly, providing

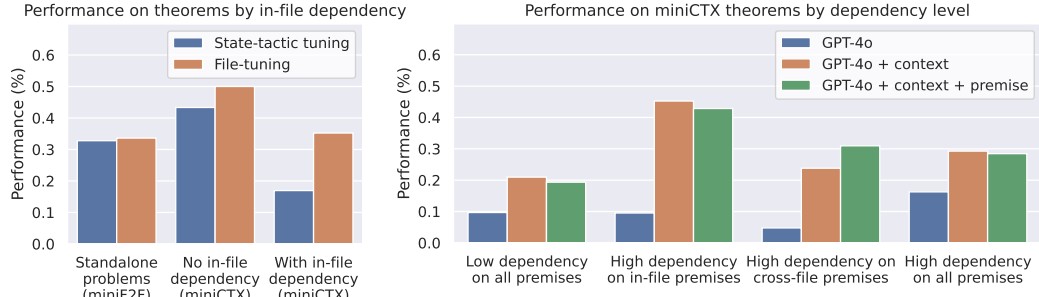

Figure 4: Model performance by dependency on premises. For each theorem in `miniCTX`, we record as metadata whether its human-written proof depends on other definitions or theorems in the same file ("in-file") or in other files ("cross-file"), and test the performance of baselines on each type.

the preceding file context, which includes definitions and lemmas, to GPT-4o results in dramatic improvement compared to using just the proof state (27.08% vs. 11.72%). These findings highlight the importance of providing models with rich contextual information beyond the immediate proof state, also demonstrating that `miniCTX` is able to measure this ability of context-dependent proving.

**Premise selection improves performance on high cross-file dependency splits.** The results in Table 3 indicate that premise selection has a mixed impact on model performance. For the GPT-4o, premise selection improves performance on high cross-file dependency splits, such as PFR, PFR$_{cross}$, and SciLean. This suggests that premise selection helps capture the cross-file context, enabling GPT-4o to make better use of cross-file information. However, for the file-tuned model, premise selection does not consistently improve results, and even performs worse on the PFR$_{cross}$ split, which was designed to evaluate the effective use of cross-file premises. This suggests that the retrieved premises differ significantly from the in-file context. Therefore, developing methods that effectively support the integration of cross-file context (e.g., premise selection) alongside in-file context remains an interesting open research direction for improving performance on the `miniCTX` benchmark.

**Evaluation on `miniF2F`.** We evaluate baselines on `miniF2F`, a standard benchmark based on competition problems that do not require context. We use import statements as context for the file-tuned model. The file-tuned model improves very little beyond the state-tactic model (33.61% vs. 32.79%), showing that the dramatic difference in context-dependent proving abilities seen on `miniCTX` cannot be captured by `miniF2F`.

## 4.3 ANALYSIS

We analyze the baseline models on `miniCTX` further along several axes, including the kinds of contextual dependencies, the difficulty, and the content made available in the context.

**File-tuning especially helps on problems with infile dependencies.** We use the `miniCTX` metadata to categorize theorems based on their in-file dependencies. Figure 6 shows the performance of state-tactic tuned model and file-tuned model on problems with in-file dependencies compared to those without. We also show `miniF2F` as an additional reference point for problems without in-file dependencies. The file-tuned model shows a marked improvement over the state-tactic tuned model, especially in problems that have dependencies on context. We conclude that file-tuning specifically helps in the realistic setting of theorem proving with new definitions and theorems in context.

**Premise selection helps but may interfere with in-file context.** We use `miniCTX` metadata to categorize problems based on their cross-file dependencies, evaluating the impact of premise selection across the entire dataset. As shown in Figure 4, GPT-4o benefits significantly from premise selection on problems with high cross-file dependencies, showing improved performance when leveraging relevant premises from imported files. However, we also observe that premise selection can interfere with in-file context, leading to inconsistent results, particularly when the available in-file context is relatively short. This suggests that adding cross-file premises may sometimes disrupt the model's ability to focus on the in-file information. Further analysis of this interference is included in §D.3. This highlights the need for more sophisticated integration strategies that can balance both in-file and cross-file contexts effectively.

Table 4: Ablation study on different context components for theorem proving.

| Environment | Definitions | Lemma Statement | Lemma Proof | Natural Language Comments | File-tuning | GPT-4o |
|---|---|---|---|---|---|---|
| ✗ | ✗ | ✗ | ✗ | ✗ | 14.12% | 8.24% |
| ✓ | ✗ | ✗ | ✗ | ✗ | 25.88% | 2.35% |
| ✓ | ✓ | ✗ | ✗ | ✗ | 24.71% | 9.41% |
| ✓ | ✓ | ✓ | ✗ | ✗ | 27.06% | 22.35% |
| ✓ | ✓ | ✓ | ✓ | ✗ | 32.94% | **34.12%** |
| ✓ | ✓ | ✓ | ✗ | ✓ | 28.24% | 23.53% |
| ✓ | ✓ | ✓ | ✓ | ✓ | **35.29%** | 31.76% |

**Models can learn from previous proofs in the file context.** To determine the contribution of different components in the in-file context, we conducted an ablation study on the `PFR.ForMathlib.Entropy.Basic` file, which contains numerous co-related lemmas and rich natural language comments, making it an ideal candidate to investigate the influence of different context components. In this ablation, we systematically removed specific parts of the in-file context and evaluated the model's ability to generate proofs under these modified conditions. As shown in Table 4, both the file-tuned model and GPT-4o benefit from the inclusion of previous proofs in the file context. This indicates that models are capable of learning proof strategies from existing proofs in the file and effectively applying them to new problems (see §D.4 for more examples).

**Natural language comments contribute in certain settings.** Our ablation also explored the effect of natural language comments in the in-file context. Though the impact was not dramatic, comments written in natural language were found to be helpful in certain settings. In scenarios where proofs were excluded from the context, adding comments resulted in slight performance gains for both models. For the file-tuned model, these gains were further amplified when proofs were included alongside comments, demonstrating the value of combining formal context with explanatory natural language. However, for GPT-4o, the presence of comments when proofs were included led to a slight decrease in performance, suggesting that effective context selection may vary depending on the model architecture and underlying training characteristics.

**File-tuning improves across all difficulty levels and context lengths.** Finally, Appendix §D.2 shows performance on problems categorized by the length of the human-written proof (when available), which we take as a rough proxy of the problem difficulty. The file-tuned model improved on all three difficulty categories. Appendix §D.2 also shows that file-tuning had improved accuracy across context lengths, particularly for problems with longer contexts. Longer contexts may imply more dependencies, suggesting that these problems can benefit more from file-tuning.

**Models rely on common symbolic automation.** To demonstrate an additional kind of context-dependence, we perform an additional analysis on Math2001 (Macbeth, 2023), which is another Lean textbook setting.[2] In particular, the textbook code disables powerful automation tactics including `simp` and `linarith` to promote manual reasoning, akin to traditional textbook exercises. For example, Math2001 includes numerous arithmetic problems that are trivial with automation tactics (e.g., `linarith`) but are challenging for models to explicitly prove with step-by-step reasoning (e.g., via `calc`). In Table 6 we evaluate models with the automation disabled, and observe substantial performance drops, confirming the reliance on automation tactics. We also find that the state-tactic tuned model relies on `simp` for unseen definitions, making it performing similarly well to the file-tuned model on theorems that only rely on new definitions (§D.6).

## 5 DISCUSSION AND FUTURE CHALLENGES

In addition to general improvements in performance, we comment on some specific open challenges.

**Making better use of long-contexts.** Our file-tuning method simply truncates contexts to be within a token budget (1024), which can discard useful contextual information. We found gains in providing GPT-4o 8,000 tokens of context compared to not providing it context, but its absolute performance was still low. There are several possible strategies that can be explored in future work, including feeding in the entire context, retrieval, or mixtures of the two.

---

[2]See Appendix B.1 for further details on Math2001. Due to licensing we do not include it in `miniCTX`.

**Repository-level context.** We focused on evaluating in-file context in this paper. As shown in §D.1, many problems require using context outside of the current file. Although we incorporated premise selection as a means of leveraging cross-file context, our experiments indicate that it does not consistently improve performance, even on datasets with high cross-file dependencies. This suggests a need to further investigate how to better integrate premise selection with in-file context. `miniCTX` provides sufficient metadata to reconstruct the entire environment, allowing for comprehensive investigation into premise selection and other potential methods for leveraging cross-file context.

**Challenging proofs.** Using context through file tuning did not improve performance on the challenging PFR proofs. Moreover, performance is relatively low (19%) on proofs that had a human-written proof of longer than five lines (see §D.2). Proving these kinds of theorems remains an open problem.

**Working with constraints.** As shown in Table 6, model performance drops when the proof cannot use powerful automation tactics. Models have a tendency to invoke these powerful tactics, and struggle with more explicit step-by-step proofs. Improving performance in this setting of `miniCTX` is an interesting future direction.

## 6    RELATED WORK

**Formal theorem proving with language models.** GPT-*f* (Polu & Sutskever, 2020) pioneered the use of language models for theorem proving via next tactic prediction given the current proof state, a technique adopted by many subsequent methods (Jiang et al., 2021; Han et al., 2022; Lample et al., 2022; Polu et al., 2022; Welleck & Saha, 2023; Azerbayev et al., 2024). ReProver (Yang et al., 2023) conditions each generation on retrieved premises, while Draft-sketch-prove (Jiang et al., 2023) conditions each generation on an informal proof. Baldur (First et al., 2023) fine-tunes a model with 50 lines of the preceding file content as context, but unlike file-tuning trains the model to generate a full proof without proof states. More broadly, machine learning for formal theorem proving is an active research area; see Lu et al. (2023); Li et al. (2024) for surveys.

**Theorem proving data extraction.** Several tools extract training data from interactive theorem provers, including CoqGym (Yang & Deng, 2019) for Coq, PISA (Jiang et al., 2021) for Isabelle, LeanStep (Han et al., 2022) for Lean 3, and LeanDojo (Yang et al., 2023) for Lean 3 and 4. Recently, `lean-training-data` (Morrison, 2023) provides tools for extracting proof states and other information using Lean 4 metaprogramming, which we anecdotally found to be easiest to modify and fastest among Lean 4 data extraction tools. Our NTP-TOOLKIT adds 3 new tools on top of this code, along with a pipeline for running on any Lean projects and instruction tuning.

**Theorem proving benchmarks.** Theorem proving methods are typically evaluated in two settings: (1) standalone competition problems (Zheng et al., 2022) or textbook (Azerbayev et al., 2023) problems; (2) holding out theorems from a mathematical library that the model is trained on, such as Mathlib for Lean (Han et al., 2022; Polu et al., 2022; Yang et al., 2023) or the Archive of Formal Proofs for Isabelle (Jiang et al., 2021; First et al., 2023). The first does not test the use of context, while the second tests only theorem-level generalization. `miniCTX` is designed to test the use of context as well as theorem-level, context-level, and project-level generalization across several mathematical domains.

**Premise selection.** Premise selection is an extensively studied class of methods for handling context in theorem proving. These methods retrieve useful lemmas from previously proved results, to help prove the current theorem. Many recent premise retrievers embed theorems and premises to a common space, and then taking a similarity measure (Yang et al., 2023; Mikuła et al., 2023), or use a classifier to determine if a premise is relevant (Irving et al., 2016; Han et al., 2022). Graph2Tac (Blaauwbroek et al., 2024) proposes an online learning method that can learn from both proofs and premises in a new context, which is particularly relevant to our work since it tackles project-level generalization. However, such methods cannot combine with more advanced pre-training-based methods without risking data contamination during evaluation. `miniCTX` aims to fill this gap by our temporal split.

## 7    CONCLUSION

We studied the realistic setting of proving theorems that depend on new information and project constraints, and formulated an evaluation framework for testing generalization using real Lean projects. We built `miniCTX`, and found that the predominant method for training neural theorem provers fails to enable context dependent proving. Our file tuning method provides a strong starting point for the new challenges opened by our investigation into theorem proving with context.

ACKNOWLEDGEMENTS

Sean Welleck thanks Convergent Research, the Lean FRO, and the OpenAI Researcher Access Program for their support.

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

# Appendix

## A  MINICTX EXAMPLES

Here we give some examples of the miniCTX and its sources to illustrate the format of the data and how and why we collect certain theorems.

### A.1  EXAMPLE ENTRY

An entry in the miniCTX dataset consists of the theorem statement, preceding file contents, and metadata information. For example, given the following theorem s_eq_pow_two in context:

```
import Mathlib.Data.Real.Basic

/-!
# Square function
We define the squaring function `s : ℝ → ℝ` to be `s x := x * x`.
-/

def s (x : ℝ) : ℝ := x * x

lemma s_eq_pow_two {x : ℝ} : s x = x ^ 2 := by
  rw [s, pow_two]
```

We collect its data into `miniCTX`, formatted in JSON as follows:

```
{
  # Preceding file content
  "srcContext": "import␣Mathlib.Data.Real.Basic\\n\\n/-!\\n#␣Square␣function\\nWe␣
      define␣the␣squaring␣function␣`s␣:␣\\u211d␣\\u2192␣\\u211d`␣to␣be␣`s␣x␣:=␣x␣*␣
      x`.\\n-/\\n\\ndef␣s␣(x␣:␣\\u211d)␣:␣\\u211d␣:=␣x␣*␣x\\n\\n",

  # Theorem statement
  "theoremStatement": "lemma␣s_eq_pow_two␣{x␣:␣\\u211d}␣:␣s␣x␣=␣x␣^␣2",

  # Fully qualified theorem name
  "theoremName": "s_eq_pow_two",

  # Temporal metadata
  "fileCreated": "(git␣commit)",
  "theoremCreated": "(git␣commit)",

  # Source metadata
  "file": "MyProject/Square.lean",
  "module": "MyProject.Square",
  "positionMetadata": {
    # Line number the theorem is on
    "lineInFile": 10,
    # Number of tokens before the theorem
    "tokenPositionInFile": 152,
    # Number of premises (definitions, theorems) before the theorem
    "theoremPositionInFile": 1
  },

  # Dependency metadata
  "dependencyMetadata": {
    # Number of definitions or lemmas defined in this file that the theorem uses
    "inFilePremises": true,
    "numInFilePremises": 1,
    # Number of definitions or lemmas defined in this repository that the theorem
        uses (including in-file ones)
    "repositoryPremises": true,
    "numRepositoryPremises": 1,
    # Number of total premises (in file, repository, or otherwise)
    "numPremises": 2,
    # Modules imported in the current file
    "importedModules": ["Mathlib.Data.Real.Basic", ...]
  },

  # Proof metadata
  "proofMetadata": {
    "hasProof": true,
    "proof": "by\n␣␣rw␣[s,␣pow_two]",
    "proofType": "tactic",
    "proofLengthLines": 2,
    "proofLengthTokens": 20
  }
}
```

In additional to individual entries, we also record the version (git commit) of the repository.

## A.2 PRIME NUMBER THEOREM EXAMPLE

We collect theorems from the `Rectangle.lean` file in PrimeNumberTheoremAnd. The following excerpt from `Rectangle.lean` demonstrates the scenario that often arises in a theorem proving environment where context is critical to producing a proof:

```
import Mathlib.Analysis.Complex.CauchyIntegral
import Mathlib.Analysis.Complex.Convex
```

```
open Complex Set Topology

open scoped Interval

variable {z w : ℂ} {c : ℝ}

/-%%
\begin{definition}\label{Rectangle}\lean{Rectangle}\leanok
A Rectangle has corners $z$ and $w \in \C$.
\end{definition}
%%-/
/-- A `Rectangle` has corners `z` and `w`. -/
def Rectangle (z w : ℂ) : Set ℂ := [[z.re, w.re]] ×ℂ [[z.im, w.im]]

namespace Rectangle

lemma symm : Rectangle z w = Rectangle w z := by
  simp [Rectangle, uIcc_comm]

lemma symm_re : Rectangle (w.re + z.im * I) (z.re + w.im * I) = Rectangle z w := by
  simp [Rectangle, uIcc_comm]
```

When proving the final lemma `symm_re`, a model can benefit much from the preceding file contents, which include (1) the existing imports from Mathlib, `variable` declarations, and open namespaces that provide a syntactic context for this theorem, (2) the new definition `Rectangle` in the context, which the model has not seen in training, (3) natural language and LaTeX documentation of the file and `Rectangle` definition, (4) the analogous (in this case identical) proof of the preceding theorem `symm`. We demonstrate that performance on `Rectangle.lean` is indeed much higher when preceding file contents are given as context to a model.

For future data added to `miniCTX` that specifically test the preceding file contents as context, we will ensure it is standalone like `Rectangle.lean`, i.e. it does not import any other unseen files from the same repository, so the preceding file contents already contain all important information relevant to the proof.

## B  ADDITIONAL DATASETS

In addition to problems in `miniCTX`, we also evaluated other datasets that are not included due to copyright reasons.

### B.1  MATH2001

Math2001 (Macbeth, 2023) contains the Lean code for the book *The Mechanics of Proof* by Heather Macbeth, an introductory text on mathematical theorem proving with accompanying Lean code. Each chapter of The Mechanics of Proof covers an introductory topic and walks through how to write the associated mathematics in Lean, along with exercises. The topics include proofs by calculation, proofs with structure, parity and divisibility, logic, induction, number theory, functions, sets, and relations. A unique aspect of Math2001 is that it disables common Lean automation for pedagogical purposes. For example, a student must write out an equality proof in detail, with each step justified. It also defines new tactics and definitions separate from the common Lean libraries. Typically a file in the textbook will show examples of such proofs, followed by exercises for a student to complete. We can view this as a form of contextual adaptation: a model must prove the theorem according to the constraints of the textbook. Math2001 has 41 files that include examples and exercises. We selected 1 to 2 theorems from each file (depending on the length of the file), for a total of 50 theorems. Of these, 31 have no proof in the Math2001 repository, hence testing theorem-level generalization.

**Context-aware models surpass state-based models**    Table 5 shows the performance comparison of different models. Both the GPT-4o model, which includes context in the input, and the file-tuned

| Models | Math2001 |
|---|---|
| GPT-4o (full proof) | 11.76% |
| GPT-4o (+ context) | 43.13% |
| State-tactic prompting | 31.37% |
| State-tactic tuning | 27.45% |
| File tuning | 41.18% |

Table 5: Performance comparison of different models on Math2001.

| Automation | File (%) | State-tactic (%) |
|---|---|---|
| Enabled | 41.18 | 11.76 |
| Disabled | 27.45 | 7.84 |

Table 6: Performance on the Math2001 split with and without access to standard automation.

model perform significantly better than the other models. This demonstrates the importance of context information in context-dependent textbook-style problems.

**Models rely on common symbolic automation.**   The Math2001 split originally disables powerful automation tactics including `simp` and `nlinarith` to promote manual reasoning, akin to traditional textbook exercises. In Table 6 we evaluate models with the automation disabled, and observe substantial performance drops, confirming a heavy reliance of current models on these automation tactics. An examination of the training corpus further revealed a general dependency on automated tactics within real Lean projects, indicating that our models have learned to rely on these tactics.

## C   NTP-TOOLKIT AND FILE-TUNING DETAILS

### C.1   DATA EXTRACTION

NTP-TOOLKIT contains a general-purpose data extraction tool that extracts examples from an arbitrary Lean 4 repository and formats them into examples that can be used to compile `miniCTX`, as well as for language-model fine-tuning. The tool is implemented in Lean based on Kim Morrison's `lean-training-data`.

Specifically, NTP-TOOLKIT takes in a configuration file with one or more Lean repositories specified. Each repository is transformed into *next-tactic* and *full proof* examples stored in JSON Lines files. The next-tactic data is suitable for making file-tuning examples of the form (context, state, next-tactic):

```
{
  "state": # tactic state ,
  "nextTactic": # pretty-printed next tactic,
  "srcUpToTactic": # source code in the file up to the tactic invocation,
  "decl": # declaration without proof (e.g., statement of a theorem),
  "declUpToTactic": # source code in the declaration up to the tactic invocation,
  "declId": # unique identifier of the declaration
}
```

The full proof data is suitable for making evaluation examples of the form (context, theorem, proof):

```
{
  "srcUpToDecl": # source code in the file up to the declaration,
  "decl": # declaration without proof (e.g., statement of a theorem),
  "declId": # unique identifier of the declaration,
  "proof": # proof
}
```

Full proof data is also suitable for training a model to directly generate a full proof, and NTP-TOOLKIT also provides Lean source with proof states interleaved, both of which we do not explore in this work.

## C.2 INPUT-OUTPUT FORMATTING.

Below we show the inputs and outputs for file-tuning and state-tactic tuning. In the paper we refer to the natural language description at the beginning of the input as an "instruction", and refer to a set of inputs and outputs as described below as "instruction-tuning data".

### C.2.1 FILE TUNING.

Given an example containing a state, next-tactic, and preceding file contents (`srcUpToTactic`), the data is formatted as:

*Input*:

```
/- You are proving a theorem in Lean 4.
You are given the following information:
- The file contents up to the current tactic, inside [CTX]...[/CTX]
- The current proof state, inside [STATE]...[/STATE]

Your task is to generate the next tactic in the proof.
Put the next tactic inside [TAC]...[/TAC]
-/
[CTX]
{srcUpToTactic}
[/CTX]
[STATE]
{state}
[/STATE]
[TAC]
```

*Output*:

```
{nextTactic}
[/TAC]
```

### C.2.2 STATE-TACTIC TUNING.

Given an example containing a state and next-tactic, the data is formatted as:

*Input*:

```
/- You are proving a theorem in Lean 4.
You are given the following information:
- The current proof state, inside [STATE]...[/STATE]

Your task is to generate the next tactic in the proof.
Put the next tactic inside [TAC]...[/TAC]
-/
[STATE]
{state}
[/STATE]
[TAC]
```

*Output*:

```
{nextTactic}
[/TAC]
```

### C.2.3 GPT-4O PROMPT

For full proof generation task with only theorem statement, we use the following prompt:

Your task is to generate complete proofs for problems stated in Lean4. You may use any tactics available in Mathlib, but no additional context, definitions, or theorems from the problem's file will be provided. Focus on crafting proofs using general knowledge and techniques applicable in Lean4. Here are some examples:

```
lemma deriv_scale {f : CS (n + 1) E} : (f.scale R).deriv = R⁻¹ ·
    f.deriv.scale R := by
  ext v ; by_cases hR : R = 0 <;> simp [hR, scale]
  · simp [deriv, smul] ; exact deriv_const _ _
  · exact ((f.hasDerivAt (R⁻¹ · v)).scomp v (by simpa using (hasDerivAt_id
    v).const_smul R⁻¹)).deriv

theorem mul_dvd_mul_left (a : α) (h : b | c) : a * b | a * c := by
  obtain ⟨d, rfl⟩ := h
  use d
  rw [mul_assoc]

/- Now here is your exercise. There is no need to restate the problem. If
    needed, think through the proof using comments. -/
```

{theorem statement}

For full proof generation task with additional infile context, we use the following prompt:

Your task is to generate complete proofs for problems stated in Lean4. For each problem, you will be provided with the context from the file in which the theorem is stated. This context includes useful external libraries, along with important definitions and theorems that are relevant to the proof. You are encouraged to use any tactics, definitions, lemmas, or theorems defined within this context to construct your proof. Please pay careful attention to indentation and formatting to ensure that the proof adheres to Lean4 syntax standards. Here are some examples:

```
#Context:
import Mathlib.Analysis.Calculus.Deriv.Support
import Mathlib.Analysis.Distribution.SchwartzSpace
import Mathlib.Order.Filter.ZeroAndBoundedAtFilter

open Real Complex MeasureTheory Filter Topology BoundedContinuousFunction
    SchwartzMap  BigOperators

variable {E : Type*} [NormedAddCommGroup E] [NormedSpace ℝ E] {{n : ℕ}}

@[ext] structure CS (n : ℕ) (E : Type*) [NormedAddCommGroup E] [NormedSpace
    ℝ E] where
  toFun : ℝ → E
  h1 : ContDiff ℝ n toFun
  h2 : HasCompactSupport toFun

noncomputable def scale (g : CS n E) (R : ℝ) : CS n E := by
  by_cases h : R = 0
  · exact ⟨0, contDiff_const, by simp [HasCompactSupport, tsupport]⟩
  · refine ⟨fun x => funscale g R x, ?_, ?_⟩
    · exact g.h1.comp (contDiff_const.smul contDiff_id)
    · exact g.h2.comp_smul (inv_ne_zero h)

/- Truncated -/

/- Now here is your exercise. There is no need to restate the problem. If
    needed, think through the proof using comments. -/
#Context:
{}

#Problem:
{}

{theorem statement}
```

# D    ADDITIONAL RESULTS AND ANALYSIS

We present additional analysis on the composition of `miniCTX` and additional quantitative and qualitative evaluation of performance on `miniCTX`. All tests in this section are done on `miniCTX`-test.

## D.1    DEPENDENCY DISTRIBUTION

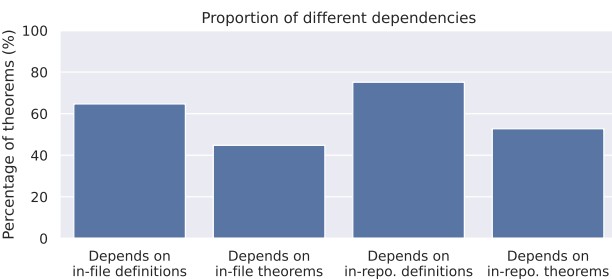

Figure 5: Percentage of different dependencies in the human-written proof of theorems in `miniCTX`.

## D.2    PERFORMANCE BY PROOF LENGTH AND CONTEXT LENGTH

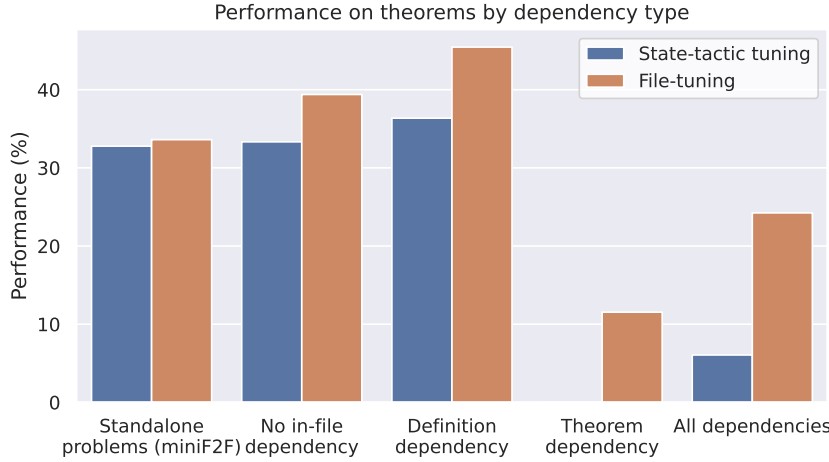

Figure 6: Performance by dependency type. For each theorem in `miniCTX`, we record as metadata whether its human-written proof depends on other definitions or theorems in the same file, and test the performance of baselines on each type. File-tuned models substantially outperform state-tactic tuned models on theorems with definition and/or theorem dependencies.

## D.3    INTERFERENCE BETWEEN IN-FILE CONTEXT AND RETRIEVED PREMISES

In our experiments, we attempted to supply both in-file context (in the form of preceding code) and premise context (in the form of retrieved premises) to GPT-4o for proving a theorem. In Figure 8, we present an analysis of the impact of the length of retrieved premises on the resulting proof success rate.

**Longer retrieved premises hurt performance.**    The results indicate that problems with a lower premise-to-context length ratio tend to have higher success rates. Specifically, successful problems often feature relatively shorter premises as proportion of the full context length. This suggests that models are better able to utilize and focus on relevant in-file context when the cross-file premises are proportionally smaller. Conversely, when the length of the premises becomes relatively large

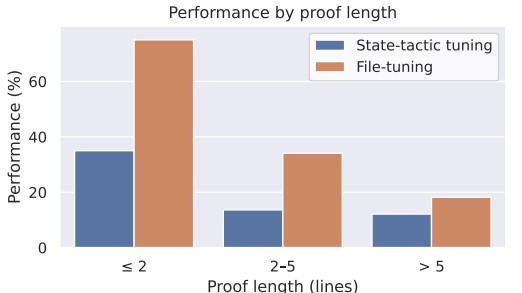 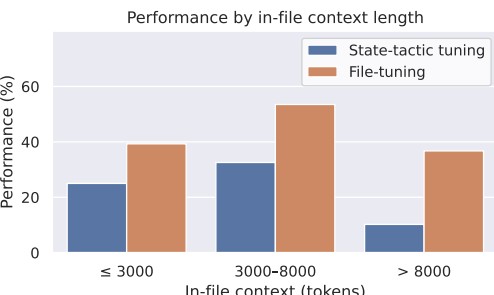

Figure 7: Performance of two baselines on different difficulty levels and context lengths, as measured by the length of human-written proof in lines and the size of the preceding file contents in tokens. File-tuning substantially improves theorem-proving abilities across all cases, but especially when the theorem is easier and the context is longer.

compared to the full context, it may overwhelm or distract the model, reducing its ability to effectively utilize the in-file information. This finding highlights the importance of ensuring a balanced integration of premises with the in-file context to maintain model focus and improve proof generation performance.

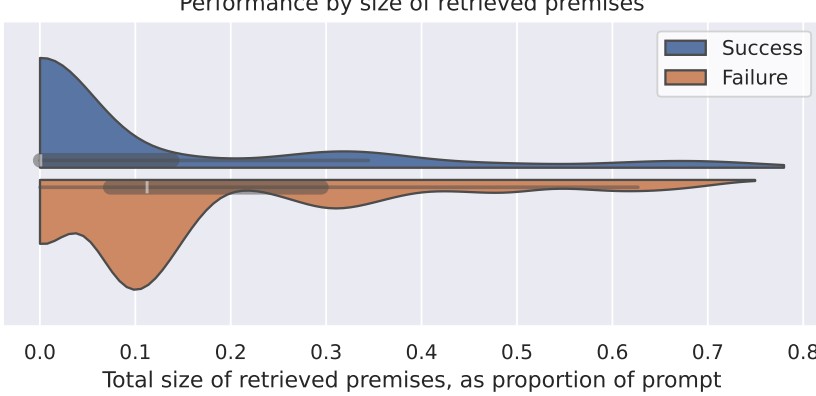

Figure 8: Impact of length of retrieved premises on GPT-4o model performance. A higher premise-to-context length ratio is correlated with lower success rates, suggesting that too much premise in context overwhelms the model.

## D.4 EXAMPLE OF LEARNING PROOFS FROM CONTEXT

The file-tuned model is able to utilize the proofs in the context. Here is an example of the model making minimal modification to the proofs from the context:

*Input*:

```
...

theorem Set.left_not_mem_uIoo {a b : ℝ} : a ∉ Set.uIoo a b :=
  fun ⟨h1, h2⟩ ↦ (left_lt_sup.mp h2) (le_of_not_le (inf_lt_left.mp h1))

theorem Set.right_not_mem_uIoo {a b : ℝ} : b ∉ Set.uIoo a b := by
```

*Output*:

```
    rintro ⟨h1, h2⟩
    exact (right_lt_sup.mp h2) (le_of_not_le (inf_lt_right.mp h1))
```

## D.5 EXAMPLE OF USING THEOREMS FROM CONTEXT

The file-tuned model is able to utilize the stated theorems in the context. Here is an example of the model using the previously defined theorem in the proof:

*Input*:

```
...
lemma RectSubRect {x₀ x₁ x₂ x₃ y₀ y₁ y₂ y₃ : ℝ} (x₀_le_x₁ : x₀ ≤ x₁)
    (x₁_le_x₂ : x₁ ≤ x₂)
    (x₂_le_x₃ : x₂ ≤ x₃) (y₀_le_y₁ : y₀ ≤ y₁) (y₁_le_y₂ : y₁ ≤ y₂) (y₂_le_y₃
    : y₂ ≤ y₃) :
    Rectangle (x₁ + y₁ * I) (x₂ + y₂ * I) ⊆ Rectangle (x₀ + y₀ * I) (x₃ + y₃ *
    I) := by
  rw [rect_subset_iff, mem_Rect, mem_Rect]
  refine ⟨⟨?_, ?_, ?_, ?_⟩, ?_, ?_, ?_, ?_⟩
  all_goals simpa using by linarith

lemma RectSubRect' {z₀ z₁ z₂ z₃ : ℂ} (x₀_le_x₁ : z₀.re ≤ z₁.re) (x₁_le_x₂ :
    z₁.re ≤ z₂.re)
    (x₂_le_x₃ : z₂.re ≤ z₃.re) (y₀_le_y₁ : z₀.im ≤ z₁.im) (y₁_le_y₂ : z₁.im ≤
    z₂.im)
    (y₂_le_y₃ : z₂.im ≤ z₃.im) :
    Rectangle z₁ z₂ ⊆ Rectangle z₀ z₃ := by
```

*Output*:

```
simpa using RectSubRect x₀_le_x₁ x₁_le_x₂ x₂_le_x₃ y₀_le_y₁ y₁_le_y₂
    y₂_le_y₃
```

## D.6 EXAMPLE OF USING UNSEEN DEFINITIONS

The state-tactic tuned model is able to utilize the unseen definitions that appear in the proof state. In the following example `Set.uIoo` is a newly defined definition, which is never seen for state-tactic tuned model:

*Input*:

```
theorem uIoo_of_le {α : Type*} [Lattice α] {a b : α} (h : a ≤ b) : Set.uIoo
    a b = Ioo a b := by

[STATE]
1 goal
zw: ℂ
c: ℝ
α: Type u_1
inst: Lattice α
ab: α
h: a ≤ b
⊢ uIoo a b = Ioo a b
[/STATE]
```

*Output*:

```
simp [uIoo, h]
```

This leads the state-tactic tuned model to perform similarly well as the file-tuned model on problems that only rely on new definitions, and not new theorems (see Figure 6).

## E  DATASET HOSTING AND MAINTENANCE

miniCTX is released on HuggingFace: `l3lab/miniCTX`, distributed under the Apache 2.0 license. Data extraction tool NTP-TOOLKIT is released on GitHub: `cmu-l3/ntp-toolkit`, under the MIT license. We note that the underlying data for the individual splits of miniCTX are also released under the Apache 2.0 license. We include the licensing information in the dataset repository. We plan to regularly update and maintain the dataset to include examples from new projects. For information about future updates such as miniCTX-v2, please refer to our project page: `https://cmu-l3.github.io/minictx`.

## F  NTP-TOOLKIT GUIDELINE

We introduced NTP-TOOLKIT in §3.4. With the NTP-TOOLKIT, users can extract and annotate new theorems and proofs from any valid Lean project, in miniCTX format. The extracted data can be used either as updates to miniCTX, or as training data (for which we also provide instruction tuning utilities). We also develop a lightweight evaluation framework for easy evaluation on miniCTX.

### F.1  PRELIMINARY

The evaluation code relies heavily on the Lean REPL (Lean Prover Community, 2024), which operates within the project environment. Therefore, it is essential that the project builds without any errors. Additionally, the version of Lean used in the project should match the version supported by the REPL. While the Lean REPL supports versions $\geq 4.3.0$, for the best experience with data extraction and evaluation, we recommend evaluating projects that use Lean version 4.7.0 or higher (all miniCTX theorems are in 4.7.0). We plan to continuously update NTP-TOOLKIT to support newer versions.

### F.2  USING THE NTP-TOOLKIT

The NTP-TOOLKIT is designed to easily extract and annotate theorem proving data from Lean projects, by simply providing the project URL. To use the NTP-TOOLKIT for data extraction, follow these steps:

1. Installation: Clone the NTP-TOOLKIT repository from GitHub to your local machine. Currently, to use NTP-TOOLKIT for Lean version 4.7, switch to the `lean-v4.7.0` branch; for 4.8.0 or above, use the `main` branch. Ensure that you have the required dependencies installed, as listed in the repository's README file.

2. Configuration: Supply GitHub URL, commit hash, and root modules of your Lean project in a JSON configuration file. Make sure that your project is using a compatible version of Lean. NTP-TOOLKIT will extract data from all modules imported by the root modules.

3. Data extraction: Run the data extraction script provided by the toolkit. Specify the `--full_proof_training_data` and `--premises` options to extract miniCTX-style data, which will be stored in an `minictx.jsonl` output file. Specify the `--declarations` option to additionally extract the premises in each module, for premise retrieval. The `full_proof_training_data` outputs can be additionally used for fine tuning (assuming the extracted data is dated before the current temporal split of miniCTX).

For detailed commands and additional options, please refer to the README file in the NTP-TOOLKIT repository.

### F.3 miniCTX EVALUATION

We provide a comprehensive evaluation pipeline in the `miniCTX-eval` repository, supporting both tactic-prediction and full-proof generation tasks. Users should place the extracted JSONL file from the NTP-TOOLKIT into the data folder. To run an evaluation task, execute the task script by specifying the dataset path, the corresponding project path, and the path to the Lean REPL. This setup ensures that the evaluation is conducted within the correct environment and with the necessary data inputs.

## G DATA CONTAMINATION IN EXISTING BENCHMARKS

One of the contributions of miniCTX is the temporal split: using NTP-TOOLKIT, miniCTX only includes theorems created after a certain cut-off date. This ensures LLMs trained before this date have not seen problems in miniCTX. We claimed that existing benchmarks face significant risks of data contamination, and here we provide some evidence to this claim.

With regards to contamination, existing benchmarks can be largely categorized as either extracted from a standard mathematical library (e.g. Mathlib), or curated as competition-style problems. For the former category, evaluation benchmarks include Mathlib test problems extracted by LeanDojo (Yang et al., 2023) or PACT (Han et al., 2022), as well as datasets like CoqGym (Yang & Deng, 2019) and PISA (Jiang et al., 2021) in other formal languages. They are inherently at risk of contamination, because they are statically sourced from projects on GitHub, many of which have existed for years (e.g. Mathlib has existed since 2017). On the other hand, virtually all modern language models are pre-trained on public GitHub code, and have therefore seen the proofs. This partially invalidates any evaluation results using these benchmarks.

On the other hand, we have empirically found many solutions of datasets like miniF2F (Zheng et al., 2022) online. The original repository of miniF2F already contains full solutions to 72 of the 244 test problems (in Lean 3, which is directly translatable to Lean 4)[3]. Moreover, of the remaining problems, 12 IMO problems have full solutions in Mathlib[4] and Compfiles (a catalog of Lean proofs of competition problems)[5], at the time of writing. In total, at least 84 of the 244 problems in miniF2F-test, including 12 of the 19 IMO problems, have full solutions on GitHub. This may be an inherent issue of small competition-style benchmarks, as the theorem-proving community itself has significant interest in formalizing competition problems and solutions. The scarcity of problems from competitions like IMO each year, as well as the bias to only formalizing some categories (geometry problems are often not formalized due to difficulty) also contribute to this issue. Whether temporal split and other mitigation methods can be applied to competition-style benchmarks may also be an interesting future direction of research.

---

[3] https://github.com/openai/miniF2F

[4] https://github.com/leanprover-community/mathlib4/tree/master/Archive/Imo

[5] https://github.com/dwrensha/compfiles

