# OpenReview forum: "miniCTX: Neural Theorem Proving with (Long-)Contexts"
_ICLR.cc/2025/Conference — ICLR 2025 Oral_

### Official Review · Reviewer_jrzz · 2024-10-21

**Soundness:** 4
**Presentation:** 3
**Contribution:** 4
**Rating:** 8
**Confidence:** 4

**Summary:**

This work propose a new formal theorm proving dataset that tests a model's ability to prove formal mathematical theorems that depend on new context that is not seen during training. This work also reports the baseline results of the miniCTX dataset.

**Strengths:**

* This work provides a new dataset for the theorem proving, which tests the model's ability to prove formal mathematical theorems that depend on new context that is not seen during training. And the ability to use context is not captured by previous benchmarks such as miniF2F.
* The work provides baseline results on miniCTX, telling the difficulty of the proposed miniCTX.
* The work provides detailed analysis on the experiments.
* The work provides details of the miniCTX dataset and samples in the Appendix, which makes it easy to understand what the miniCTX looks like.

**Weaknesses:**

* For the Table 3, how do you test the File tuning Model result on miniF2F, as the miniF2F only has the formal statement? You may: 1) only give the formal statement for File tuning Model and get the result; 2) Or, you provide addtional context information by any way?;
* For Table 3, you do not report the result of GPT-4o (full proof), could you explain why? Previous works (DSP, LEGO-Prover or Lyra) all reports the results GPT-4 on miniF2F.

**Questions:**

Please check the weakness

---

> ### Author Response · Authors · 2024-11-21
>
> Thank you for your review and your feedback on our paper. We address your questions and concerns below.
> > 1. For the Table 3, how do you test the File tuning Model result on miniF2F, as the miniF2F only has the formal statement? You may: 1) only give the formal statement for File tuning Model and get the result; 2) Or, you provide addtional context information by any way?
>
> Thank you for pointing this out! We provide only the imported modules as context when evaluating miniF2F. In the comparison between state-tactic tuning and file tuning on miniF2F, the inputs are almost identical. This experiment is designed to demonstrate that the file-tuning method maintains the same level of reasoning ability as traditional methods when context is limited or absent. We have updated our paper to state this.
>
> > 2. For Table 3, you do not report the result of GPT-4o (full proof), could you explain why? Previous works (DSP, LEGO-Prover or Lyra) all reports the results GPT-4 on miniF2F.
>
> We initially did not report the GPT-4o result on miniF2F, because the focus of our work is on the $\texttt{miniCTX}$ benchmark and context-aware setting. However, we agree it is better to include them for completeness of evaluation, and comparison to $\texttt{miniCTX}$ and previous works. Using the same prompt in a pass@8 setup, GPT-4o achieves 13.52% accuracy on the miniF2F test split. We have updated Table 3 to include GPT-4o performance on miniF2F in our paper.

---

> > ### Comment · Reviewer_jrzz · 2024-11-21
> > **Response to Authors**
> >
> > Dear Authors,
> >
> > Thank you very much for your precious response. The response has addressed my concerns. Also, after reading the rebuttal and comments from other reviewers, I decided to increase the score from 6 to 8 and the confidence from 3 to 4.

---

### Official Review · Reviewer_uXMR · 2024-11-02

**Soundness:** 3
**Presentation:** 3
**Contribution:** 3
**Rating:** 8
**Confidence:** 4

**Summary:**

This paper introduces a new benchmark called miniCTX for evaluating a model's context-dependent theorem proving abilities, i.e., a model's ability to maximize Expectation_{(theorem, context) ~ repository} Expectation_{proof ~ model(. | theorem, context )} [theorem is valid proof in context]. In-file context includes the source code that precedes the theorem in the file while cross-context includes the in-file context and relevant premises from imported modules. The authors argue that this is a more realistic evaluation of a model's theorem proving abilities. Additionally, the paper introduces NTP-toolkit, a tool for constructing/extending miniCTX with future repositories and running baseline evaluations.

**Strengths:**

- The paper introduces a useful tool (NTP-toolkit) and benchmark (miniCTX) for evaluating a model's theorem proving capabilities. The Python Lean REPL, provided that it is performant, is another useful component that will enable researchers to more easily evaluate models.
- A benchmark of context-dependent theorem proving and a model that performs well on it could potentially be useful for assisting humans using Lean for formal verification.
- The idea of file tuning, i.e., training model(tactic | theorem, context) is novel.

**Weaknesses:**

- The differences between benchmarks shown in Table 1 is, in my perspective, superficial. For example, what fundamental reason is there from preventing other theorem-proving extraction tools from extracting a timestamp or saving the file name that a theorem is extracted from?
- There seem to be a few missing experiments. First, since the temporal split is emphasized, is there any empirical evidence to support the failure of other benchmarks to handle this properly? Second, for file-tuning, have other LLMs / those fine-tuned on Mathlib been tested with file-tuning to see if file-tuning works across different models? Third, In Table 4, I'd like to see context = environment + definition + lemma statement but without natural language comments. This way, we can see just how much having the lemma proof affects file-tuning to better support the claim that models can learn from previous proofs in context.
-  While the paper offers empirical evidence and some analysis, there isn't much in the form of hypotheses to guide the design of the benchmark or experiments. As an example hypothesis, while it might not be surprising that model(tactic | theorem, context) > model(tactic | theorem) since we condition on more information, we hypothesize that the gains come mostly from context = previous proofs since this offers the appropriate context as to when an automated tactic can be most beneficially applied. Such hypotheses would help guide and focus the experiments, and offer more insight into the success and failure cases of neural theorem proving with LLMs.

**Questions:**

- The weaknesses section contains questions that I'd be curious to hear the answers to.
- Why doesn't the cross-file context include the source code of imported modules as opposed to just the premises? Is this just because of context-length limitations?

---

> ### Author Response · Authors · 2024-11-21
> **Rebuttal by Authors (1/2)**
>
> Thank you for your review and your feedback on our paper.
>
> > 1. The differences between benchmarks shown in Table 1 is, in my perspective, superficial. For example, what fundamental reason is there from preventing other theorem-proving extraction tools from extracting a timestamp or saving the file name that a theorem is extracted from?
>
> $\texttt{miniCTX}$ differs from previous theorem-proving extraction tools in its motivation. We aim to test the theorem-proving ability of LLMs in a real-world, context-dependent setting that is safe from potential data contamination. This is a different motivation than benchmarks that test models without contextual information. $\texttt{miniCTX}$ is the first benchmark that incorporates full real-world context for theorem proving. As demonstrated in our context ablation analysis (Table 4), different components contribute to the performance differently, showing the importance of the full context. Second, our motivation is different from benchmarks extracted from a static snapshot of a project (e.g. LeanDojo [1]) that the model has also trained on. While in principle new snapshots might be re-extracted or carefully split based on time, these ideas have not been realized in a benchmark. Table 1 thereby reflects these differences, showing the position of $\texttt{miniCTX}$ among relevant work.
>
> Our NTP-TOOLKIT is also the first unified, automated, and simple framework for extracting context-level data and creation timestamps from any Lean repository. It lowers the cost of extraction for future evaluations. We are also the first benchmark to provide a comprehensive evaluation on the importance of contextual information.
>
> > 2. First, since the temporal split is emphasized, is there any empirical evidence to support the failure of other benchmarks to handle this properly?
>
> Yes! All public static benchmarks in Lean such as LeanDojo’s extracted Mathlib [1] and miniF2F [2] are at risk of data contamination. For LeanDojo, any model (e.g. internLM [3], DeepSeek-Prover [4]) that has pre-trained on public GitHub data, including Mathlib, has seen the proofs in the eval and test splits of LeanDojo’s benchmark (note that Mathlib4 has large parts copied from its predecessor Mathlib3, which was created in 2017).
>
> For another example in competition-style datasets, a significant portion of problems in miniF2F-test have solutions available on GitHub. In the original miniF2F repository [2], there are 72 test problems already stated with complete proofs. Additionally, the following IMO problems in miniF2F-test: 1959 P1, 1960 P2, 1977 P6, 1982 P1, 2001 P6, 2019 P1 are all fully formalized in Lean inside Mathlib’s Archive/Imo library [5]; 1962 P2, 1963 P5, 1964 P2, 1965 P2, 1968 P5, 1981 P6 are all fully formalized in Lean in Compfiles [6]. So at least 84/244 theorems in miniF2F-test, including 12/19 IMO problems, have full solutions on GitHub.
>
> We have added a new Appendix G (referenced on L274) to support this claim, in order to provide a clearer motivation for the temporal split in $\texttt{miniCTX}$.
>
> [1] Kaiyu Yang, Aidan Swope, Alex Gu, Rahul Chalamala, Peiyang Song, Shixing Yu, Saad Godil, Ryan Prenger, and Anima Anandkumar. LeanDojo: Theorem proving with retrieval-augmented language models. In Neural Information Processing Systems (NeurIPS), 2023.
>
> [2] Kunhao Zheng, Jesse Michael Han, and Stanislas Polu. miniF2F: a cross-system benchmark for formal olympiad-level mathematics. In International Conference on Learning Representations, 2022. URL https://openreview.net/forum?id=9ZPegFuFTFv.
>
> [3] Wu, Zijian, et al. "InternLM2. 5-StepProver: Advancing Automated Theorem Proving via Expert Iteration on Large-Scale LEAN Problems." arXiv preprint arXiv:2410.15700 (2024).
>
> [4] Xin, Huajian, et al. "DeepSeek-Prover-V1. 5: Harnessing Proof Assistant Feedback for Reinforcement Learning and Monte-Carlo Tree Search." arXiv preprint arXiv:2408.08152 (2024).
>
> [5] https://github.com/leanprover-community/mathlib4/tree/master/Archive/Imo
>
> [6] https://github.com/dwrensha/compfiles
>
> > 3. Second, for file-tuning, have other LLMs / those fine-tuned on Mathlib been tested with file-tuning to see if file-tuning works across different models?
>
> Thank you for this suggestion! We are currently file-tuning a Llama 3 8B model and evaluating it on $\texttt{miniCTX}$, which we will hopefully complete before the discussion period ends (currently, our computational resources are limited).
>
> The idea behind file-tuning is to provide the full file context to the model during proof generation. While we have demonstrated the effectiveness of file-tuning on DeepSeek-Coder-1.3B, we have also observed substantial performance improvements when using few-shot prompting with context for GPT-4o and Llemma-7B. Although these larger models were not fine-tuned using the file-tuning dataset, the underlying principle of leveraging full context remains consistent. So there is reasonable belief that file-tuning works across different models.

---

> > ### Author Response · Authors · 2024-11-21
> > **Rebuttal by Authors (2/2)**
> >
> > > 4. Third, In Table 4, I'd like to see context = environment + definition + lemma statement but without natural language comments. This way, we can see just how much having the lemma proof affects file-tuning to better support the claim that models can learn from previous proofs in context.
> >
> > This experiment setting is already presented on row 4 of Table 4. Indeed, making proofs of previous lemmas available in the context significantly increases performance, by 5.88% and 11.77% absolute for file-tuning and GPT-4o respectively (as seen in Table 4, row 4 vs 5).
> >
> > > 5. While the paper offers empirical evidence and some analysis, there isn't much in the form of hypotheses to guide the design of the benchmark or experiments. As an example hypothesis, while it might not be surprising that model(tactic I theorem, context) > model(tactic I theorem) since we condition on more information, we hypothesize that the gains come mostly from context = previous proofs since this offers the appropriate context as to when an automated tactic can be most beneficially applied. Such hypotheses would help guide and focus the experiments, and offer more insight into the success and failure cases of neural theorem proving with LLMs.
> >
> > There are a few overarching hypotheses. First, a guiding hypothesis is that incorporating full context significantly improves theorem proving. As none of the existing formal theorem proving benchmarks incorporate contextual information, our $\texttt{miniCTX}$ benchmark therefore collects this information, and evaluates performance of LLMs that better incorporate such information. We confirm that providing contextual information significantly boosts model performance. We also hypothesized that a model could have roughly the same performance on a competition-style benchmark, but have much better performance on context-dependent proving. We confirmed this experimentally.
> >
> >
> > To further investigate how context contributes to performance gains, we tested three more hypotheses: (1) preceding theorem statements help, (2) preceding proofs help, and (3) natural language comments help. As detailed in our ablation experiments (Table 4, Section 4.3), we evaluated these hypotheses by isolating and testing different context components. For example, one observation is that providing preceding proofs to the model increases file-tuning model performance by about 6%, and GPT-4o performance by about 10% (Table 4, row 4 vs 5 and row 6 vs 7). This supports the claim that the model can learn from previous proofs in the context and apply them to the current proof, which is something not provided or evaluated by previous benchmarks that have no contextual information or only consider premise selection.
> >
> > > 6. Why doesn't the cross-file context include the source code of imported modules as opposed to just the premises? Is this just because of context-length limitations?
> >
> > Evaluating performance with putting all source code of imported modules is indeed an interesting future direction of this work. We have not done it here because of context-length limitations of our models, and for a fair comparison between different models. Based on our estimates, the average cross-file context for each problem exceeds 368k tokens (excluding mathlib), which far exceeds the capacity of current models and our computational resources. Even the full list of premises is too lengthy, so we rely on premise selection to represent cross-file context in a more manageable way.
> >
> > On the other hand, an interesting observation is that as the length of retrieved premises increases, the performance actually degrades (Appendix D.3). We postulate that putting too much irrelevant information in context can adversely impact model performance, and plan to investigate this in future work.

---

> > > ### Comment · Reviewer_uXMR · 2024-11-21
> > >
> > > Thank you for your thorough response and efforts in updating the paper to address them.
> > > Apologies that I missed rows 4/5 which addresses my primary questions about file-tuning and the effect of lemma proofs.
> > > A more direct discussion of the hypotheses included in these responses and the thought process behind them would be my only other suggestion for improvement.
> > > You have addressed my concerns and I will update my score to reflect this.
> > > I look forward to trying out MiniCTX.

---

### Official Review · Reviewer_gDKA · 2024-11-02

**Soundness:** 3
**Presentation:** 4
**Contribution:** 4
**Rating:** 8
**Confidence:** 3

**Summary:**

This paper introduces $\texttt{miniCTX}$, a benchmark of 384 problems with context-rich information, e.g., in-file definitions and lemmas, and evaluates the model's theorem-proving ability under new context, which claims to be closer to the real-world scenario when researchers develop the formal repository. The paper further includes the NTP-TOOLKIT that the authors use for data extraction. The authors provide several baseline experiments, including the common practice of state-tactic tuning and prompting methods (which works for standalone problems as presented in $\texttt{miniF2F}$) and file-tuning that works under such a context-rich setup.

**Strengths:**

1. The $\texttt{miniCTX}$ benchmark fills the gap in the current community: it expands current benchmarks by addressing the limitations of standalone theorem proving and enabling evaluation in real-world scenarios where context is critical.

2. I appreciate the authors' commitment to automatically updating the benchmark and maintaining a temporal split to mitigate the data contamination, given that most LLMs nowadays crawl GitHub and train on it.

3. The authors present details for constructing the benchmark and the sources. The addition of the $\texttt{miniCTX}$ and the NTP-TOOLKIT will be valuable assets to the community. The authors also present solid baselines in Table 3 using inference-time methods with GPT-4o and fine-tuning methods using 1.3b model. An ablation of providing different contexts is also presented in Table 4 to show the source of gain from each context component.

**Weaknesses:**

1. $\texttt{miniCTX}$ does not have a valid/test set separation. Though it's not inherently an issue for a benchmark, separating a valid set could make the benchmark less gameable under Goodhart's law.

2. It seems that some problems in $\texttt{miniCTX}$ are with contexts that could be easily "in-lined" and transformed into context-less problems. For example, the example shown in Appendix A.1: one could easily in-line the square function $s$ definition into the lemma s\_eq\_pow\_two and make the statement to be $x * x = x ^ 2$ from $s \ x = x ^ 2$. The same in-line transformation seems to be also applied to the example in Appendix A.2 by inlining the Rectangle definition. It would be great for the author to assess how many problems in $\texttt{miniCTX}$ could be transformed into context-free problems under certain efforts.

**Questions:**

1. Is the 1.3b model in L349 the same as the DeepSeek Coder 1.3b in L343?

2. What is the "Environment" in Table 4? Does it stand for the import statements / open namespace statements or something else?

---

> ### Author Response · Authors · 2024-11-21
>
> Thank you for your review and your feedback on our paper. We address your questions and concerns below.
>
> > 1. $\texttt{miniCTX}$ does not have a valid/test set separation. Though it's not inherently an issue for a benchmark, separating a valid set could make the benchmark less gameable under Goodhart's law.
>
> We thank the reviewer for this suggestion. Indeed, we are planning to add a validation split to $\texttt{miniCTX}$. While the extraction process using our toolkit only takes a few minutes, evaluating all models on the new split will require additional time. We plan to include the validation split as well as corresponding evaluation in the final camera-ready version.
>
> > 2. It seems that some problems in $\texttt{miniCTX}$ are with contexts that could be easily "in-lined" and transformed into context-less problems. For example, the example shown in Appendix A.1: one could easily in-line the square function $s$ definition into the lemma s_eq_pow_two and make the statement to be $x * x = x^2$ from $s x = x^2$. The same in-line transformation seems to be also applied to the example in Appendix A.2 by inlining the Rectangle definition. It would be great for the author to assess how many problems in $\texttt{miniCTX}$ could be transformed into context-free problems under certain efforts.
>
> The in-line transformation is indeed applicable to the examples in Appendix A.1 and A.2. It more generally holds for theorems whose statement involves only new Lean `def`s or `abbrev`s, essentially amounting to calling using the tactic `unfold` in the first line of the proof. This transforms a context-dependent problem into a context-free problem (while losing information such as previous proofs and lemmas).
>
> From a manual inspection, many theorems in $\texttt{miniCTX}$ can technically be transformed in this way, by recursively unfolding definitions. We note that this cannot be applied to `class`, `structure`, or `inductive` type definitions, or certain recursive definitions: e.g. there is no easy way to rewrite `Nat.add a b` to a form without `Nat.add`.
>
> However, we do not believe unfolding definitions is a valid approach to proving context-dependent problems. For example, a later theorem in Rectangle.lean [1] is: `square_neg (p : ℂ) (c : ℝ) : Square p (-c) = Square p c`. While we can transform this by definition of `Square p c = Rectangle (-c - c * I + p) (c + c * I + p)` and then recursively `Square p c = [[(-c - c * I + p).re, (c + c * I + p).re]] ×ℂ [[(-c - c * I + p).im, (c + c * I + p).im]]`, the actual proof benefits from the fact that a `Square` is a `Rectangle` and the lemma `Rectangle.symm` (proved in the example given in Appendix A.2). In other words, even if the theorem statement itself can be transformed to a context-free problem, the actual proof might benefit from the lemmas in the context, which are furthermore stated in an unexpanded format. It is up to the prover whether the definitions should be unfolded, and whether lemmas in context should be used. Moreover, the transformed statement can be exceedingly long, if there are multiple levels of definition (consider let `s x := x * x`, `t x := s (s x)`, `u x := t (t x)`, and then `u e` for some long `e` becomes `(((e * e) * (e * e)) * ((e * e) * (e * e))) * (((e * e) * (e * e)) * ((e * e) * (e * e)))`; long chains of definitions are common in e.g. the HepLean split).
>
> [1] https://github.com/AlexKontorovich/PrimeNumberTheoremAnd/blob/main/PrimeNumberTheoremAnd/Rectangle.lean
>
>
> > 3. Is the 1.3b model in L349 the same as the DeepSeek Coder 1.3b in L343?
>
> Yes. We thank the reviewer for pointing this out, and we have revised the relevant part in our paper.
>
> > 4. What is the "Environment" in Table 4? Does it stand for the import statements / open namespace statements or something else?
>
> Yes, the “'Environment” includes imports, open namespace statements, variable declarations, and other contextual elements necessary for the theorem's setup. To distinguish it from definitions and lemmas, the environment does not introduce new information.

---

> ### Comment · Reviewer_gDKA · 2024-11-21
> **Response to Author**
>
> I appreciate the author's response. My questions and concerns are properly addressed. I put my confidence and trust for the authors for their commitment of maintaining a temporal split and I believe miniCTX, along with the Tookit, could be valuable assets to the community.
>
> I will maintain my score and recommend an acceptance.

---

### Author Response · Authors · 2024-11-21
**Overall Response**

## General Response
We thank the reviewers for valuable and encouraging feedback. We are grateful to all three reviewers for recognizing the important contributions of a benchmark for context-dependent theorem proving in real-world scenarios, and the baseline evaluations presented on $\texttt{miniCTX}$. We appreciate reviewers gDKA and uXMR for acknowledging the utility of our data extraction tool NTP-TOOLKIT for the research community, and reviewers gDKA and jrzz for recognizing the details of dataset construction and analysis. We respond to each reviewer separately below. We have also revised our paper to incorporate the feedback.

We first summarize the motivation and main contributions of $\texttt{miniCTX}$.

In recent years, large language models have been applied to completing verified mathematical proofs in interactive theorem provers [1, 2]. However, existing benchmarks fail to evaluate this ability of LLMs—competition benchmarks such as miniF2F [3] do not incorporate the rich contextual information seen in real-world theorem proving settings, while benchmarks based on a library such as Mathlib [4, 5, 6] at most include a subset of the available context, such as premises. State-of-the-art LLM-based provers are also trained without incorporating the full context [7, 8]. Moreover, all existing benchmarks face significant risks of data contamination, undermining the accuracy of their evaluations.

$\texttt{miniCTX}$ is the first benchmark that evaluates formal theorem proving in a context-aware, real-world setting. Specifically, we:
- (1) provide the $\texttt{miniCTX}$ benchmark sourced from real-world projects, incorporating all contextual information necessary for a proof;
- (2) provide ntp-toolkit, which automatically extracts relevant theorems from Lean projects, while ensuring no data contamination by using a temporal split;
- (3) evaluate several baselines on $\texttt{miniCTX}$, demonstrating that a simple but effective method, file-tuning, significantly helps real-world theorem proving.

We hope $\texttt{miniCTX}$ will enable richer and more accurate evaluations of theorem-proving abilities of LLMs.

## Revisions to the paper
* L349: “a 1.3b model” → “DeepSeek-Coder-1.3b”
* Added a new Appendix G (referenced on L274) showing data contamination issues in other benchmarks, to support the importance of temporal split
* Added evaluation setup details on miniF2F (L407)
* Added performance of GPT-4o on miniF2F (Table 3)

## Ongoing experiments
* To further strengthen our findings and verify if file-tuning generalizes across different models, we are currently evaluating file-tuning on Llama-3.1 8B.
* Evaluating baselines on a new test split

## Thank you
Thank you again for taking the time to review our paper, and we sincerely hope that you take these discussions into consideration when re-evaluating the work. We have also responded in more detail to your specific concerns below.

## Reference

[1] Sean Welleck and Rahul Saha. LLMSTEP: LLM proofstep suggestions in Lean. arXiv preprint arXiv:2310.18457, 2023.

[2] Peiyang Song, Kaiyu Yang, and Anima Anandkumar. Towards large language models as copilots for theorem proving in Lean. arXiv preprint arXiv: Arxiv-2404.12534, 2024.

[3] Kunhao Zheng, Jesse Michael Han, and Stanislas Polu. miniF2F: a cross-system benchmark for formal olympiad-level mathematics. In International Conference on Learning Representations, 2022. URL https://openreview.net/forum?id=9ZPegFuFTFv.

[4] Jesse Michael Han, Jason Rute, Yuhuai Wu, Edward Ayers, and Stanislas Polu. Proof artifact cotraining for theorem proving with language models. In International Conference on Learning Representations, 2022. URL https://openreview.net/forum?id=rpxJc9j04U.

[5] Kaiyu Yang, Aidan Swope, Alex Gu, Rahul Chalamala, Peiyang Song, Shixing Yu, Saad Godil, Ryan Prenger, and Anima Anandkumar. LeanDojo: Theorem proving with retrieval-augmented language models. In Neural Information Processing Systems (NeurIPS), 2023.

[6] Kaiyu Yang and Jia Deng. Learning to prove theorems via interacting with proof assistants, 2019.

[7] Huaiyuan Ying, Shuo Zhang, Linyang Li, Zhejian Zhou, Yunfan Shao, Zhaoye Fei, Yichuan Ma, Jiawei Hong, Kuikun Liu, Ziyi Wang, et al. InternLM-Math: Open math large language models toward verifiable reasoning. arXiv preprint arXiv:2402.06332, 2024.

[8] Huajian Xin, Z. Z. Ren, Junxiao Song, Zhihong Shao, Wanjia Zhao, Haocheng Wang, Bo Liu, Liyue Zhang, Xuan Lu, Qiushi Du, Wenjun Gao, Qihao Zhu, Dejian Yang, Zhibin Gou, Z. F. Wu, Fuli Luo, and Chong Ruan. DeepSeek-Prover-v1.5: Harnessing proof assistant feedback for reinforcement learning and monte-carlo tree search. arXiv preprint arXiv:2408.08152, 2024. URL https://arxiv.org/abs/2408.08152.

---

> ### Author Response · Authors · 2024-12-04
> **Additional Results**
>
> We sincerely thank the reviewers for their thoughtful and constructive feedback, as well as for recognizing the significance of our contributions.
>
> We are pleased to share the results of our ongoing experiments with the Llama-3.1-8B model. Specifically, we tested both state-tactic tuning and file-tuning on this model, and its performance on the $\texttt{miniCTX}$ benchmark improved from **18.75%** to **23.70%**, demonstrating the effectiveness of the file-tuning method. However, the improvement is less pronounced compared to the DeepSeek-Coder model. We hypothesize that this is primarily due to the Llama-3.1 model lacking pretraining on formal math related data compared to other models we are testing. The full result is shown in the following table:
>
> | Method                                  | miniF2F-test | Prime  | PFR   | PFR\_cross | Mathlib | HTPI   | HEP    | SciLean | Avg.  |
> |---------------------------------------|--------------|--------|-------|------------|---------|--------|--------|---------|-------|
> | **GPT-4o (full proof)**           | 13.52        | 7.06   | 1.85  | 6.98       | 14.00   | 13.33  | 31.15  | 6.52    | 11.72 |
> | &emsp; + context                  | —         | 31.76  | 5.56  | 34.88      | 26.00   | **17.78** | 49.18  | 17.39   | 27.08 |
> | &emsp; + context + premise | —         | 29.41  | 7.41  | 39.53      | —    | 15.56  | 44.26  | 21.74   | 26.82 |
> | **LLEMMA-7B (State-tactic prompting)**           | 28.28        | 20.00  | 5.56  | 0.00       | 16.00   | 0.00   | 31.15  | 19.57   | 14.58 |
> | **DeepSeek-Coder-1.3B (State-tactic tuning)** | 32.79        | 17.65  | 5.56  | 0.00       | 22.00   | 11.11  | 52.46  | 19.57   | 19.53 |
> | **DeepSeek-Coder-1.3B (File tuning)**  | **33.61**    | 40.00  | 5.56  | **44.19**  | **34.00** | 15.56  | **60.66** | **45.65** | **35.94** |
> | &emsp; + premise                 | —         | **42.35** | **11.11** | 16.28   | —    | 8.89   | 50.82  | 32.61   | 30.21 |
> | **Llama-3.1-8B (State-tactic tuning)**     | —              | 20.00        | 1.96   | 0.00  | 22.00      | 4.44    | 49.18  | 23.91  | 18.75 |
> | **Llama-3.1-8B (File tuning)**    | —               | 38.82        | 3.70   | 20.93 | 22.00      | 6.67    | 40.98  | 17.39  | 23.70 |
>
> We thank the reviewers once again for their time and effort in providing detailed feedback and raising thoughtful questions. We hope that $\texttt{miniCTX}$, as a more comprehensive benchmark under realistic settings, will serve as a valuable resource for the formal mathematics community, inspiring further exploration into the potential of neural theorem provers.

---

### Meta-Review · Area_Chair_NCAm · 2024-12-21

**Metareview:**

This paper introduces miniCTX, a novel benchmark for evaluating neural theorem-proving models in real-world, context-rich scenarios, alongside NTP-toolkit, an automated tool for data extraction and annotation. The propositions are well-motivated, addressing the limitations of existing benchmarks (e.g., miniF2F, which focuses on isolated problems) and emphasizing the importance of context in theorem proving.

The reviewers unanimously appreciated the benchmark’s contribution to advancing formal mathematics research, particularly the attention to mitigating data contamination through temporal splits. Baseline evaluations and ablation studies were also thorough and insightful, showcasing the effectiveness of context-aware approaches like file-tuning.

There were some shared concerns among reviewers, including the absence of a validation split, the potential for simplifying problems into context-free forms, and incomplete experiments on additional models. However, the rebuttal addressed these issues to a reasonable extent, with the authors committing to further improvements (e.g., adding a validation split and additional experiments).

Overall, I agree with the reviewers that this paper offers significant contributions to the field. The authors are encouraged to incorporate the additional evaluations and clarifications discussed during the rebuttal phase when preparing the final version of the paper.

**Additional Comments On Reviewer Discussion:**

During the rebuttal period, reviewers raised concerns about the lack of a validation split, potential simplification of problems into context-free forms, incomplete experiments (e.g., testing file-tuning across more models), and the novelty of miniCTX compared to existing benchmarks. The authors addressed these points by clarifying that many problems inherently rely on contextual dependencies, presenting preliminary results for ongoing experiments, and providing evidence of contamination issues in other benchmarks.

---

### Decision · Program_Chairs · 2025-01-22

Accept (Oral)